# Food and Non-Food-Related Behavior across Settings in Children with Prader–Willi Syndrome

**DOI:** 10.3390/genes11020204

**Published:** 2020-02-17

**Authors:** Marie G. Gantz, Sara M. Andrews, Anne C. Wheeler

**Affiliations:** RTI International, Research Triangle Park, NC 27709, USA; sandrews@rti.org (S.M.A.); acwheeler@rti.org (A.C.W.)

**Keywords:** Prader–Willi syndrome, food-related behavior, childhood

## Abstract

This study sought to describe food- and non-food-related behaviors of children aged 3 to 18 years with Prader–Willi syndrome (PWS) in home and school settings, as assessed by 86 parents and 63 teachers using 7 subscales of the Global Assessment of Individual’s Behavior (GAIB). General Behavior Problem, Non-Food-Related Behavior Problem, and Non-Food-Related Obsessive Speech and Compulsive Behavior (OS/CB) scores did not differ significantly between parent and teacher reports. Food-Related Behavior Problem scores were higher in parent versus teacher reports when the mother had less than a college education (difference of 13.6 points, 95% Confidence Interval (CI) 5.1 to 22). Parents assigned higher Food-Related OS/CB scores than teachers (difference of 5.7 points, 95% CI 2.4 to 9.0). Although teachers reported fewer Food-Related OS/CB, they scored overall OS/CB higher for interfering with daily activities compared with parents (difference of 0.9 points, 95% CI 0.4 to 1.4). Understanding how behaviors manifest in home and school settings, and how they vary with socio-demographic and patient characteristics can help inform strategies to reduce behavior problems and improve outcomes.

## 1. Introduction

Prader–Willi syndrome (PWS), a genetic disorder caused by a lack of expression on paternal chromosome 15 (15q11-q13) [1,2], manifests as a result of three causes: paternal deletion, maternal uniparental disomy (UPD), or imprinting defect. The most common subtype is the paternal deletion, which accounts for approximately 70 percent of all PWS cases [2,3]. Paternal deletions are classified as Type I (TI) or Type II (TII), based on the size of the deletion [2,4]. UPD accounts for approximately 25 percent of PWS cases, and imprinting defects are the cause of approximately 5 percent of cases [2,5]. 

Although the most notable clinical feature of PWS is hyperphagia, which can begin in early childhood and can lead to obesity and related health issues [2,5,6], behavioral challenges are also common. The behavioral phenotype of PWS generally includes food-seeking behaviors, tantrums, repetitive speech, obsessions, compulsions, and self-injurious behaviors, such as skin picking, as well as internalizing problems such as feelings of negative self-worth, withdrawal, and sadness [5,7,8,9,10]. 

Behavioral challenges vary across age and PWS subtype. In infancy, there are more physical than behavioral concerns, including hypotonia, feeding difficulties, failure to thrive, hypogonadism, lethargy, and decreased interest in feeding [11,12]. Early childhood often brings a significant increase in both externalizing (tantrums, aggression, stealing, etc.) and internalizing (anxiety and depression, skin picking, etc.) behaviors [10,13]. Intellectual disabilities (IDs) and social difficulties become more apparent as children progress through the school-age years. Most individuals with PWS have mild to moderate ID and, even among those with higher IQs, learning and social problems are common [14,15,16]. Many children with PWS have difficulties relating to peers and often prefer to be with older or younger groups of children [7]. In addition, children with PWS may withdraw into more solitary activities, such as word searches and jigsaw puzzles, rather than engaging in social activities [7]. During early and late adulthood, severe psychiatric illness, such as depression and affective psychosis may develop, especially in those with UPD [17,18,19,20]. Maladaptive and compulsive behaviors that began in childhood, such as overeating, hoarding, and tantrums may be elevated into middle adulthood but have been reported to greatly diminish in older adulthood [10,21]. 

While physical, cognitive, and behavioral concerns are almost universally reported in PWS, there is some evidence to suggest the manifestation of these concerns differs by subtype. Compared to individuals with UPD, individuals with the deletion subtype have been found to demonstrate more compulsive behavior [9,22,23,24], which has been implicated in both social and academic challenges. Within the deletion subtypes, some have reported no significant difference in behavior between TI and TII deletion [21,25], and those who have found a difference between the two differ in their report on which deletion subtype exhibits more severe compulsive behavior [23,26].

Due the extreme hyperphagia characteristic of PWS, many of the behaviors typically seen in individuals with PWS are food related (i.e., seeking and hoarding food, impulsivity, repetitive requests for food). However, many of the behaviors described by parents include non-food-related concerns as well, such as hoarding non-food items, needing to ask or tell something, ordering and arranging objects, and repeating rituals [27,28]. 

As most studies examining PWS phenotypic expression have depended on parent report, there is a dearth of information regarding specific differences in food- and non-food-related behaviors at school compared to the home environment. Although, to date, no studies have specifically compared behaviors at home versus at school, several aspects of the PWS phenotype suggest there may be significant differences in behaviors in these two settings. For example, research suggests that behavioral outbursts in children with PWS often occur in conjunction with unexpected changes in settings or routines, which is likely to pose a problem at school [29,30,31]. There is some evidence that children with TII deletions especially may struggle with compulsive behaviors that relate to specific academic areas while those with TI deletions may have more compulsions related to social activities and grooming [24], suggesting behavioral challenges may differ between subtypes at school as well as at home. Furthermore, individuals with PWS may experience more social problems at school as a result of increased exposure to and relationships with peers [7,24,32]. Conversely, the structured setting and increase in distractions may result in a reduction of food-related behavior problems compared to the home setting where there may be more access to food and where most meals are likely to take place.

The goals of this study were to explore in greater detail the food- and non-food-related behavior patterns of children with PWS as assessed by parents for the home setting and by teachers for the school setting. Associations between behavior scores and participant characteristics including PWS genetic subtype were also explored. 

## 2. Materials and Methods

### 2.1. Participants

Children with PWS between the ages of 3 and 18 years participated in this study as part of a larger project exploring development, learning, and behavioral profiles in children with PWS. Parents or primary caregivers of the children were invited to participate through one of four means: (1) the Prader–Willi Clinic located at a university medical center; (2) invitations on list-servs and websites serving families and educators of children with PWS; (3) through national organizations for individuals and families of individuals with PWS; (4) and through invitational letters distributed at the annual Prader–Willi Syndrome Association conference. Parents distributed the questionnaires to their child’s primary teacher, who returned the completed forms directly to the researchers. 

### 2.2. Procedures

The study was approved by the institutional review board of the University of North Carolina, Chapel Hill, UNC IRB # 05-2572. Parents of all subjects gave their informed consent for their child’s inclusion in the study, and children provided assent, as appropriate. After receiving verbal consent, parents were sent packets, which included parent and teacher consent forms, release forms for the school and doctor, parent and teacher rating forms, and several rating scales including a demographic form and the Global Assessment of Individual’s Behaviors [33]. The packet also included letters explaining the study, directions for completing forms and rating scales, and self-addressed and stamped envelopes for returning the documents. Parents were asked to provide teachers with the letter and forms to be completed and mailed individually for their child’s assessment. For a subsample of children, Stanford–Binet Abbreviated Brief IQ scores [34] were obtained during a research-based visit to the university clinic. PWS diagnosis and subtype were reported by parents and confirmed via genetic report. For most participants, these reports included follow-up testing to determine subtype; however, for 17 participants, only the methylation test confirming PWS was available.

### 2.3. Measures

#### 2.3.1. Global Assessment of Individual’s Behavior (GAIB)

The Global Assessment of Individual’s Behavior (GAIB) [33] is a rating instrument that is designed to assess and identify behavior problems typically associated with PWS. The GAIB-PWS was adapted from the Nisonger Child Behavior Rating Form (CBRF), a rating scale which was normed on children and adolescents with developmental disabilities [35]. The GAIB is comprised of 7 subscales: Social Competence (10 items with values ranging from 0 = ”Not True” to 3 = ”Always True”), General Behavior Problems (40 items with values ranging from 0 = ”Not a Problem” to 3 = ”Major Problem”), Food-Related Behavior Problems (24 items with values ranging from 0 = ”Not a Problem” to 3 = ”Major Problem”), Non-Food-Related Behavior Problems (24 items with values ranging from 0 = ”Not a Problem” to 3 = ”Major Problem”), Food-Related Obsessive Speech and Compulsive Behavior (OS/CB) (16 items with values ranging from 0 = ”Not a Problem” to 3 = ”Major Problem”), Non-food-Related OS/CB (16 items with values ranging from 0 = ”Not a Problem” to 3 = ”Major Problem”), and Level of Interference of OS/CB with Daily Function (Interference) (4 items with values of 0 = ”No” or 1 = ”Yes”). These scales were designed primarily as a clinical measure to help assess the relative challenges across different areas of behaviors, for example, to compare food- versus non-food-related behaviors in order to optimize behavior management techniques.

Due to the highly specific nature of this rating scale (designed specifically for children with PWS), norms are not available to provide a comparison to our sample. Rather, the data provided by this measure are used to examine differences with respect to clinically relevant demographic characteristics (age, sex, race, maternal education), PWS genetic subtype, and teacher and parent report. 

#### 2.3.2. Cognitive Functioning

For participants who were seen in clinic, estimated cognitive function was obtained through administration of the Abbreviated Brief IQ (ABIQ) scale of the Stanford Binet Intelligence Scales 5th edition [34]. The ABIQ is composed of two subtests—one focused on verbal reasoning and the other on nonverbal reasoning. Although not as comprehensive as the full-scale IQ, the ABIQ provides a reliable estimate of cognitive functioning, with correlations with the full-scale IQ ranging from 0.81 to 0.85 [34,36]. The ABIQ was administered by a licensed psychologist with expertise in assessment of children with developmental disabilities. 

### 2.4. Statistical Analysis

Demographic characteristics were described as percentages or as means and standard deviations (SD) for children with GAIB assessments completed by parents, teachers, or both. Characteristics were compared between participants who had both parent and teacher assessments versus only one assessment using t tests or Chi-square tests. 

Raw scores were calculated for the 7 subscales of the GAIB by summing the scores for all component items. If a teacher did not provide an answer to the General Behavior Problems item “trouble sleeping at night,” the value was imputed as the average of the non-missing subscale items. For all subscales other than Interference, if only one question was missing (other than the “trouble sleeping” item), it was imputed to be the average value of the other items in the subscale. If more than one component question was missing an answer, the subscale was not calculated. Mean (with SD) and median (with interquartile range) scores were calculated for parent and teacher reports. 

General linear mixed models were created to compare parent and teacher scores, adjusting for patient characteristics considered to be clinically important: age, sex, race (categories of white, black, other/unknown), maternal education (categories of less than college degree, college graduate, unknown), and genetic subtype (categories of deletion, UPD/imprinting, unknown). Race, maternal education, and genetic subtype were collapsed for modeling because of small numbers in some categories. Models accounted for the within-subject correlation between parent and teacher assessments using a compound symmetry covariance structure. To assess whether there were differences between parent and teacher reports within subgroups, interactions between evaluator (parent or teacher) and other covariates were assessed, and those that were statistically significant at the α <0.05 level were retained in the final models. For scores that differed between subgroups, individual questions were examined to provide insight into which items contributed the most to those differences, but this was considered exploratory and statistical tests are not reported. 

Food-related and non-food-related scores for behavior problems and OS/CB were compared separately for parent and teacher reports using general linear mixed models that accounted for within-subject correlation between food and non-food scores using a compound symmetry covariance structure. Associations between GAIB scores and the ABIQ scale were assessed using Pearson correlation. 

Throughout the analysis, *p* values <0.05 were considered statistically significant. No adjustments were made for multiple testing; thus, the results should be interpreted cautiously. 

## 3. Results

Between February 2007 and February 2010, 149 GAIB forms were completed: 86 parent assessments, and 63 teacher assessments. GAIB assessments were returned by both parents and teachers for 49 children. One hundred children were represented—of whom, 48 were male, 82 were white, and 47 had mothers with college degrees (Table 1). The average age was 9.9 years (standard deviation 4.3). The PWS genetic subtype was known for 83 children, with 55 deletions, 27 UPD cases, and 1 imprinting error. The subsets of children with GAIB parent reports, teacher reports, or both were representative of the 100 children with any assessment, with no meaningful differences between those with one versus two raters (Appendix A). 

For parent-reported GAIB questionnaires, the amount of missing data for individual questions was relatively low, and subscales could be calculated for between 78/86 (91%) and 84/86 (98%) of returned forms (depending on the subscale) (Table 2). Teacher-completed questionnaires had higher amounts of missing data for some subscales, and the total number of calculated scores ranged from 54/63 (86%) to 61/63 (97%).

### 3.1. Associations Between GAIB Scores and Patient Characteristics

In modeling results, older age at the time of assessment was associated with a lower score for Social Competence (mean −0.4 points, 95% CI −0.6 to −0.2, for each 1 year increase in age), and higher scores for General Behavior Problems (1.2 points, 95% CI 0.5 to 1.9, for each 1 year increase in age), Food-Related Behavior Problems (1.8 points, 95% CI 1.1 to 2.4), Non-Food-Related Behavior Problems (1.6 points, 95% CI 0.9 to 2.3), and Food-Related OS/CB (0.4 points, 95% CI 0.03 to 0.8) (Table 3). 

### 3.2. Differences Between Parent- and Teacher-Reported GAIB Scores

General Behavior Problem, Non-Food-Related Behavior Problem, and Non-Food-Related OS/CB scores did not differ significantly between parent and teacher reports (Table 3). 

For Social Competence, there was an interaction between evaluator and sex (*p* value = 0.046), such that for boys, parents reported higher Social Competence than teachers (difference of 2.3 points, 95% CI 0.4 to 4.3). The individual items with the largest differences were “complied with rules or demands”, “initiated positive social interactions”, and “shared with or helped others.”

For Food-Related Behavior Problem scores, there was a statistical interaction between evaluator and maternal education (*p* value = 0.032). Further investigation revealed that parent scores were higher than teacher scores when maternal education was less than college graduate (difference of 13.6 points, 95% CI 5.1 to 22) (Table 3). The items with the largest differences were those related to crying, frustration, anger, and irritability, along with talking too much or too loudly, and difficulty transitioning activities. 

For Food-Related OS/CB scores, parents assigned higher scores than teachers (difference of 5.7 points, 95% CI 2.4 to 9.0), and girls had higher scores than boys (difference of 3.3 points, 95% CI 0.02 to 6.6) (Table 3). Differences between parent and teacher reports were largest for GAIB items related to repetitive speech and questioning, as well as “insisted on closing or opening doors or cupboards.” Differences between girls and boys were largest for excessively cleaning body parts and for hiding or hoarding objects.

Interference scores given by parents were lower than teacher reports, averaged over all children (difference of −0.9 points, 95% CI −1.4 to −0.4) (Table 3). Teachers reported more interference with social activities or regular routines, and more interference for greater than one hour per day. 

### 3.3. Food-Related Compared to Non-Food-Related GAIB Scores

Parent-reported Food-Related Behavior Problem scores were higher than Non-Food-Related scores when the mother had less than a college degree (difference of 5.8, 95% CI 0.7 to 10.8) and lower than Non-Food-Related scores when the mother was a college graduate (difference of −5.1 points, 95% CI −9.5 to −0.8) (Table 4). 

For parent-reported OS/CB scores, the difference between Food-Related and Non-Food-Related scores increased with age (difference of 0.3 points for each increase of 1 year, 95% CI 0.04 to 0.6), and there was a difference in Food-Related versus Non-Food-Related scores for boys (difference of −3.8, 95% CI −5.4 to −2.2) but not girls (Table 4). 

In teacher assessments, Food-Related scores were lower than Non-Food-Related scores for Behavior Problems (difference of −9.3 points, 95% CI −13.8 to −4.8) and for OS/CB (difference of −5.3 points, 95% CI −7.2 to −3.4) (Table 5). Averaging over Food-Related and Non-Food-Related Behavior Problem scores, teachers gave higher scores to students with UPD or imprinting PWS subtypes compared to those with deletions (difference of 7.8 points, 95% CI 0.3 to 15.4).

### 3.4. Associations Between GAIB Scores and Stanford–Binet Abbreviated Brief IQ 

In general, the ABIQ scale was negatively correlated with behavior problems. Pearson correlations were statistically significant between ABIQ and teacher-reported Food-Related Behavior Problems (correlation −0.42, *p* value = 0.017) and parent-reported General Behavior Problems (correlation −0.39, *p* value = 0.024) and Non-Food-Related Behavior Problems (correlation −0.36, *p* value = 0.044) (Table 6). 

## 4. Discussion

This study sought to expand understanding of behavioral concerns in children with PWS by including specific examination of food- versus non-food-related behaviors in different settings (home versus school). Results suggested that child age is the greatest predictor of behavior problems as assessed by parents and teachers, with older children exhibiting more behavioral challenges than younger children. This finding is consistent with previous literature describing an increase in behavior challenges in children with PWS through childhood, adolescence, and young adulthood, with a decrease occurring only once the individual reaches later adulthood [10,21]. The increasing difference with age between parent-reported scores for food- and non-food-related behavior problems is also consistent with the progression of PWS through stages of increased food interest, culminating in the onset of hyperphagia at a median age of 8 years [6]. 

Two subscales revealed sex-related differences. For boys, higher social competence was reported by parents than teachers. In the home environment, children may be more comfortable exhibiting characteristics such as socialization and helpfulness. Girls scored higher than boys for food-related obsessive speech and compulsive behaviors including hiding or hoarding. At least one other study has found that food-related behaviors differ based on both sex and genetic subtype of PWS, with less severe behavior reported in males within the UPD subtype [37]. More research studies focused on potential sex-based differences in behavioral phenotypes of PWS are needed. 

While non-food-related subscales were similar in parent and teacher reports, food-related behaviors were more likely to be noted as problematic by parents compared to teachers. These findings suggest that the food-seeking behaviors of children with PWS may be less problematic in the school setting than at home. Mothers with less education reported more food-related behavior problems such as crying, frustration, anger, difficulty transitioning, and repetitive speech than teachers, and they scored food-related behaviors as more problematic than non-food-related behaviors. In contrast, mothers with more education reported fewer food-related than non-food-related behavior problems, suggesting a complex environmental picture. It could be that mothers with lower education levels are at a disadvantage with respect to resources for managing the home environment for their child with PWS. More exploration into how families with varying education and socio-economic backgrounds manage their child’s PWS may be warranted in order to identify the best mechanisms for providing appropriate supports. 

Although teachers reported fewer food-related obsessive speech and compulsive behaviors compared to parents, teachers reported that when the behaviors occurred, they caused more interference with daily routines than was reported by parents. This is not surprising given expectations for maintaining order and adhering to schedules in the classroom, which may lead to more opportunities for obsessive speech and compulsive behaviors to be disruptive. However, it also speaks to the potential for unique challenges for children with PWS within the school setting. If obsessive speech and compulsive behaviors result in reduced ability to manage the school day, the ability to learn and appropriately interact with peers is compromised. 

Higher estimated IQ was associated with less parent-reported general and non-food-related behavior problems, and less teacher-reported food-related behavior problems. It may be that it is easier to structure the school setting and provide distractions from food for children who are less cognitively impaired. 

In this study, the only difference observed between the genetic subtypes was that teachers gave higher Behavior Problems scores to children with UPD compared to those with a deletion subtype when food- and non-food-related scores were combined in analysis. Other studies have generally found greater compulsive behavior in deletion [9,22,24] and more autistic behaviors in UPD [25,38,39]. Given the body of literature showing consistent differences in behaviors between UPD and deletion subtypes and some studies reporting differences between the two deletion subtypes [9,22,23,24,25,26], we expected to see more genetic subtype differences. However, our largest category for genetic subtypes was deletion type unknown as seen in Table 1. Thus, our findings could be associated with the small sample size in our study, with the largest genetic subtype group being comprised of deletion type unknown, or with the nature of the measure used to assess behavior. Regarding the latter, the GAIB is not designed to identify co-morbid psychiatric conditions or increased symptomology relative to other children. Rather it is designed to assess specific behaviors known to be of high prevalence in children with PWS, with specific attention to identifying which behaviors are food- and non-food-related. It could be that while general behavior problems differ in frequency and severity between genetic subtypes, when it comes to specific PWS behaviors, this is not the case. However, given our small sample size, additional research is warranted to confirm this hypothesis. Another possibility is that significant differences are more likely to be observed within subtypes in adolescence and early adulthood. Dykens et al. (2004) reported that young adults in their 20s scored highest on measures of maladaptive and compulsive behavior compared to other age groups [10]. Most studies that have reported behavioral differences between genetic subgroups have assessed young adults, with mean age in early to mid-20s [9,23,24,26], whereas the mean age of our study population was 9.9 years. 

There are several aspects of this study that limit our ability to make broad conclusions regarding the results. First, the GAIB is not a validated measure with published norms or psychometrics, which limits our ability to generalize these findings to compare with other behavioral studies. Furthermore, while the GAIB provides important information on behaviors specific to PWS, results cannot be compared to other populations. Therefore, we are unable to draw any conclusions regarding the similarities or differences in behavioral profiles of children with PWS versus those without PWS. Also, while our sample is large relative to other studies of this rare neurogenetic condition, our power to detect differences, especially between genetic subtypes, was limited by the sample size, and we did not have sufficient numbers to compare TI and TII deletions. We were also unable to collect several important variables, including degree of obesity and other co-morbid conditions, which may have provided additional insight into how and why these behaviors manifest. Additional, larger studies with more complete genetic subtype information and data on obesity, hyperphagia, and co-morbid conditions are needed to further understand food- and non-food-related behaviors in home and school settings. 

This study is the first that we know of that specifically examines food-related and non-food-related behaviors across home and school settings in children with PWS. Understanding how these behaviors manifest across different settings can help inform strategies to reduce behavior problems and improve outcomes.

## Figures and Tables

**Table 1 genes-11-00204-t001:** Characteristics of Study Participants.

Characteristic	Category	All Children (N = 100)	Parent (N = 86)	Teacher (N = 63)	Parent and Teacher (N = 49)
Age at Assessment	Mean (SD)	9.9 (4.3)	9.9 (4.4)	9.1 (3.9)	8.9 (4.0)
Male		48 (48.0%)	44 (51.2%)	30 (47.6%)	26 (53.1%)
Race	White	82 (82.0%)	74 (86.0%)	50 (79.4%)	42 (85.7%)
	Black	2 (2.0%)	2 (2.3%)	2 (3.2%)	2 (4.1%)
	Hispanic	5 (5.0%)	3 (3.5%)	4 (6.3%)	2 (4.1%)
	Bi-Racial	4 (4.0%)	3 (3.5%)	2 (3.2%)	1 (2.0%)
	Other or Unknown	7 (7.0%)	4 (4.7%)	5 (7.9%)	2 (4.1%)
Maternal Education Level	HS Graduate or Less	9 (9.0%)	8 (9.3%)	7 (11.1%)	6 (12.2%)
	Some College	25 (25.0%)	23 (26.7%)	16 (25.4%)	14 (28.6%)
	College Graduate	30 (30.0%)	27 (31.4%)	20 (31.7%)	17 (34.7%)
	Graduate Degree	17 (17.0%)	15 (17.4%)	8 (12.7%)	6 (12.2%)
	Unknown	19 (19.0%)	13 (15.1%)	12 (19.0%)	6 (12.2%)
PWS Genetic Subtype	Deletion Type I	9 (9.0%)	8 (9.3%)	7 (11.1%)	6 (12.2%)
	Deletion Type II	15 (15.0%)	15 (17.4%)	8 (12.7%)	8 (16.3%)
	Deletion Type Unknown	31 (31.0%)	25 (29.1%)	19 (30.2%)	13 (26.5%)
	UPD	27 (27.0%)	25 (29.1%)	14 (22.2%)	12 (24.5%)
	Imprinting	1 (1.0%)	1 (1.2%)	1 (1.6%)	1 (2.0%)
	Unknown	17 (17.0%)	12 (14.0%)	14 (22.2%)	9 (18.4%)

SD: Standard Deviation; HS: High School; UPD: Maternal Uniparental Disomy.

**Table 2 genes-11-00204-t002:** Raw Global Assessment of Individual’s Behavior (GAIB) Subscale Scores.

Statistic Type	Parent	Teacher
**Social Competence**		
N	84	59
Mean (SD)	17.5 (5.4)	16.9 (4.3)
Median (IQR)	18 (14, 21)	16.7 (15, 18.9)
**General Behavior Problems**		
N	82	61
Mean (SD)	25 (17.4)	25.5 (16.5)
Median (IQR)	22.5 (12, 35)	23 (15, 31.8)
**Food-Related Behavior Problems**		
N	82	61
Mean (SD)	23.6 (19.4)	15.6 (16.3)
Median (IQR)	17 (8.3, 35)	10 (1, 26)
**Non-Food-Related Behavior Problems**		
N	80	57
Mean (SD)	23.5 (16)	24.1 (16.1)
Median (IQR)	20.5 (12, 32.5)	21.5 (11.5, 32)
**Food-Related Obsessive Speech and Compulsive Behavior**		
N	78	55
Mean (SD)	11.7 (11.3)	5.8 (6.3)
Median (IQR)	7 (3, 19)	3.5 (0, 10)
**Non-Food-Related Obsessive Speech and Compulsive Behavior**		
N	81	54
Mean (SD)	14.2 (10.5)	11 (8.1)
Median (IQR)	12 (6, 21)	8 (4.5, 17)
**Level of Interference of Obsessive Speech and Compulsive Behavior with Daily Function**		
N	81	58
Mean (SD)	2 (1.6)	3 (1.5)
Median (IQR)	2 (1, 4)	4 (2, 4)

N: Number; IQR: Interquartile Range.

**Table 3 genes-11-00204-t003:** Model Results for GAIB Subscales.

Model Effect	Category	Estimate (95% CI)	*P* value
**Social Competence**			
Age at Assessment	Increase of 1 Year	−0.4 (−0.6, −0.2)	<0.001
Sex (Average Effect)	Female vs. Male	−0.5 (−2.3, 1.4)	0.6
Race	Non-White vs. White	0.1 (−2.8, 3)	0.94
Maternal Education	< College Grad vs. College Grad	0.5 (−1.5, 2.6)	0.61
PWS Genetic Type	UPD/Imprinting vs. Deletion	−0.2 (−2.4, 1.9)	0.83
Evaluator (Average Effect)	Parent vs. Teacher	0.9 (−0.4, 2.3)	0.18
Evaluator*Sex Interaction	Parent vs. Teacher (Females)	−0.5 (−2.4, 1.5)	0.65
Evaluator*Sex Interaction	Parent vs. Teacher (Males)	2.3 (0.4, 4.3)	0.018
Evaluator*Sex Interaction	Female vs. Male (Parent Assessment)	−1.9 (−4, 0.2)	0.08
Evaluator*Sex Interaction	Female vs. Male (Teacher Assessment)	0.9 (−1.6, 3.4)	0.47
**General Behavior Problems**			
Age at Assessment	Increase of 1 Year	1.2 (0.5, 1.9)	<0.001
Sex	Female vs. Male	4.3 (−1.3, 9.8)	0.13
Race	Non-White vs. White	5.9 (−3.2, 15)	0.2
Maternal Education	< College Grad vs. College Grad	2.7 (−3.5, 8.8)	0.39
PWS Genetic Type	UPD/Imprinting vs. Deletion	1.2 (−5.4, 7.7)	0.73
Evaluator	Parent vs. Teacher	−0.5 (−5.8, 4.8)	0.86
**Food-Related Behavior Problems**			
Age at Assessment	Increase of 1 Year	1.8 (1.1, 2.4)	<0.001
Sex	Female vs. Male	3.3 (−2.3, 8.9)	0.25
Race	Non-White vs. White	7.8 (−1.4, 17)	0.1
Maternal Education (Average Effect)	< College Grad vs. College Grad	4.6 (−1.7, 11)	0.15
PWS Genetic Type	UPD/Imprinting vs. Deletion	2.7 (−3.9, 9.3)	0.42
Evaluator (Average Effect)	Parent vs. Teacher	8.9 (3.2, 14.6)	0.002
Evaluator*Education Interaction	Parent vs. Teacher (Mother < College Grad)	13.6 (5.1, 22)	0.002
Evaluator*Education Interaction	Parent vs. Teacher (Mother is College Grad)	−0.3 (−7.9, 7.2)	0.93
**Non-Food-Related Behavior Problems**			
Age at Assessment	Increase of 1 Year	1.6 (0.9, 2.3)	<0.001
Sex	Female vs. Male	2 (−3.5, 7.5)	0.46
Race	Non-White vs. White	−0.3 (−9.2, 8.7)	0.95
Maternal Education	< College Grad vs. College Grad	−0.1 (−6.3, 6)	0.96
PWS Genetic Type	UPD/Imprinting vs. Deletion	3.5 (−3, 9.9)	0.29
Evaluator	Parent vs. Teacher	−2 (−6.6, 2.7)	0.4
**Food-Related Obsessive Speech and Compulsive Behavior**			
Age at Assessment	Increase of 1 Year	0.4 (0, 0.8)	0.028
Sex	Female vs. Male	3.3 (0, 6.6)	0.049
Race	Non−White vs. White	4.7 (−0.6, 10)	0.08
Maternal Education	< College Grad vs. College Grad	1 (−2.7, 4.6)	0.59
PWS Genetic Type	UPD/Imprinting vs. Deletion	−1.8 (−5.7, 2.1)	0.36
Evaluator	Parent vs. Teacher	5.7 (2.4, 9)	<0.001
**Non-Food-Related Obsessive Speech and Compulsive Behavior**			
Age at Assessment	Increase of 1 Year	0.3 (−0.1, 0.7)	0.2
Sex	Female vs. Male	1.4 (−2.2, 5)	0.43
Race	Non-White vs. White	3.4 (−2.4, 9.3)	0.25
Maternal Education	< College Grad vs. College Grad	−0.2 (−4.2, 3.8)	0.92
PWS Genetic Type	UPD/Imprinting vs. Deletion	−1.6 (−5.9, 2.7)	0.45
Evaluator	Parent vs. Teacher	3.2 (0, 6.4)	0.05
**Level of Interference of Obsessive Speech and Compulsive Behavior with Daily Function**			
Age at Assessment	Increase of 1 Year	0 (0, 0.1)	0.17
Sex	Female vs. Male	0.2 (−0.4, 0.8)	0.48
Race	Non-White vs. White	0.8 (−0.2, 1.7)	0.11
Maternal Education	< College Grad vs. College Grad	0 (−0.7, 0.6)	0.92
PWS Genetic Type	UPD/Imprinting vs. Deletion	−0.3 (−0.9, 0.4)	0.42
Evaluator	Parent vs. Teacher	−0.9 (−1.4, −0.4)	<0.001

CI: Confidence Interval; PWS: Prader–Willi syndrome.

**Table 4 genes-11-00204-t004:** Model Results Comparing Food- and Non-Food-Related Scores for Parent-Reported GAIB Subscales.

Model Effect	Category	Estimate (95% CI)	*p* Value
**Behavior**			
Age at Assessment	Increase of 1 Year	1.7 (1, 2.5)	<0.001
Sex	Female vs. Male	2.7 (−3.7, 9.1)	0.4
Race	Non-White vs. White	2.5 (−9.1, 14.1)	0.67
Maternal Education (Average Effect)	<College Grad vs. College Grad	6.2 (−0.8, 13.3)	0.08
PWS Genetic Type	UPD/Imprinting vs. Deletion	−0.3 (−7.6, 7)	0.93
Score Type (Average Effect)	Food vs. Non-Food	1.5 (−2, 4.9)	0.39
Score Type*Education Interaction ^1^	Food vs. Non-Food (Mother < College Grad)	5.8 (0.7, 10.8)	0.026
Score Type*Education Interaction ^1^	Food vs. Non-Food (Mother is College Grad)	−5.1 (−9.5, −0.8)	0.022
**Obsessive Speech and Compulsive Behavior**			
Age at Assessment (Average Effect)	Increase of 1 Year	0.2 (−0.3, 0.7)	0.45
Sex (Average Effect)	Female vs. Male	4 (−0.5, 8.6)	0.08
Race	Non-White vs. White	6.9 (−1.4, 15.3)	0.1
Maternal Education	<College Grad vs. College Grad	1.9 (−3.1, 7)	0.44
PWS Genetic Type	UPD/Imprinting vs. Deletion	−2.4 (−7.8, 3)	0.38
Score Type (Average Effect)	Food vs. Non-Food	−2.4 (−3.5, −1.3)	<0.001
Score Type*Sex Interaction ^2^	Food vs. Non-Food (Females)	−1 (−2.6, 0.5)	0.2
Score Type*Sex Interaction ^2^	Food vs. Non-Food (Males)	−3.8 (−5.4, −2.2)	<0.001
Score Type*Age at Assessment ^2^	Food vs. Non-Food (for Each 1-Year Increase in Age)	0.3 (0, 0.6)	0.022

^1^*P* value for interaction between score type (food or non-food) and maternal education was 0.005. ^2^
*P* values for interaction with score type (food or non-food) were 0.023 for age and 0.016 for sex.

**Table 5 genes-11-00204-t005:** Model Results Comparing Food- and Non-Food-Related Scores for Teacher-Reported GAIB Subscales.

Model Effect	Category	Estimate (95% CI)	*p* Value
**Behavior**			
Age at Assessment	Increase of 1 Year	1.3 (0.4, 2.2)	0.007
Sex	Female vs. Male	1.3 (−5, 7.5)	0.69
Race	Non-White vs. White	7 (−3, 16.9)	0.16
Maternal Education	<College Grad vs. College Grad	−2.5 (−9.4, 4.4)	0.47
PWS Genetic Type	UPD/Imprinting vs. Deletion	7.8 (0.3, 15.4)	0.043
Score Type	Food vs. Non-Food	−9.3 (−13.8, −4.8)	<0.001
**Obsessive Speech and Compulsive Behavior**			
Age at Assessment	Increase of 1 Year	0.3 (−0.1, 0.8)	0.17
Sex	Female vs. Male	−0.2 (−3.7, 3.3)	0.93
Race	Non-White vs. White	1.6 (−3.8, 7.1)	0.55
Maternal Education	<College Grad vs. College Grad	−1.5 (−5.5, 2.4)	0.44
PWS Genetic Type	UPD/Imprinting vs. Deletion	−0.2 (−4.6, 4.2)	0.93
Score Type	Food vs. Non-Food	−5.3 (−7.2, −3.4)	<0.001

**Table 6 genes-11-00204-t006:** Correlation between GAIB Subscales and Stanford–Binet Abbreviated Brief IQ.

Statistic Type	Parent	Teacher
**Social Competence**		
N	37	30
Pearson Correlation	0.06	0.33
*p* value	0.71	0.07
**General Behavior Problems**		
N	34	31
Pearson Correlation	−0.39	−0.16
*p* value	**0.024**	0.4
**Food-Related Behavior Problems**		
N	33	31
Pearson Correlation	−0.2	−0.42
*p* value	0.27	**0.017**
**Non-Food-Related Behavior Problems**		
N	32	28
Pearson Correlation	−0.36	−0.02
*p* value	**0.044**	0.91
**Food-Related Obsessive Speech and Compulsive Behavior**		
N	34	27
Pearson Correlation	−0.34	−0.17
*p* value	0.05	0.39
**Non-Food-Related Obsessive Speech and Compulsive Behavior**		
N	34	26
Pearson Correlation	−0.3	0.05
*p* value	0.09	0.8
**Level of Interference of Obsessive Speech and Compulsive Behavior with Daily Function**		
N	35	30
Pearson Correlation	−0.32	−0.21
*p* value	0.06	0.26

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
