# Peer review of "Food and Non-Food-Related Behavior across Settings in Children with Prader–Willi Syndrome"

_genes, 2020, doi:10.3390/genes11020204_

Round 1

Reviewer 1 Report

The manuscript describes a study examining differences between parent and teacher reports of food and non-food related behaviour problems evidenced by children with Prader-Willi syndrome.  Such contextual differences in the evidence of behaviour problems is a useful and under-researched area.  The manuscript is well written, the method well chosen to address the study objectives, and the interpretation of results provided is appropriate.  However, at present the rationale around some of the analytical decisions is unclear.

Specifically, the introduction makes a strong case for examining differences in teacher and parent reports.  However, there is no rationale provided for why specific demographic subgroups of people with PWS would differ in different ways in terms of associated parent versus teacher reports.  Since the primary results focus on interactions between evaluator and demographic variables, a strong rationale for why such differences based on demographic subgroup would be expected is important.  This rationale then also impacts downstream on the analysis decisions and appropriate p-value thresholds (see also below).

The statistical validity of the analysis methods applied given the small sample size but large number of demographic variables examined with very few cases of some categories is currently not addressed in the manuscript.  A rationale for why this method of analysis was deemed appropriate, and for the specific analytical decisions taken (e.g. p-value thresholds) seems extremely important.

Minor points:

1.      Please add information on outcome variables derived from the GAIB

2.      Please explain in the methods section how the demographic data used in the analysis were ascertained 

3.      In table 1, the n is missing from the parent and teacher column, please add this.

4.      In table 5, please make significant correlations clear using bold font.

Reviewer 2 Report

Gantz et al. clarified the difference of food and nonfood-related behavior of PWS individuals in home and school settings by scoring using GAIB and ABIQ. Their findings are informative for the parents and the teachers taking care of PWS individuals to reduce behavioral problems and their stress.

The methodology is straightforward, and the results are clearly presented.

This is a well written, interesting, and useful contribution, which I think is entirely suitable for publication in 'Genes'.

However, this manuscript can be accepted only after the revision of this paper considering the comments below.

Major comments

In my opinion, the results section could be shortened. Because the estimate, 95% CI and p-value for each variable are described in tables and result section. The sentences in the results section are too much detailed and could be simplified.

In line 159, the authors mentioned the genetic causes of 83 individuals enrolled in this study. Were the remaining 17 cases diagnosed molecularly by methylation analysis? Or diagnosed clinically?

Is there any data related to obesity? If there is, do the authors find any relationship between the scores and degree of obesity?

Minor comments

In line 122, does 'SB5' stand for?

Round 2

Reviewer 2 Report

This second version of the manuscript has been much improved. The results section has been revised to more succinct style. 

I think this manuscript is acceptable.

Author Response

We thank the reviewer for the comments.